# GENERATING HUMAN MOTION VIDEOS USING A CASCADED TEXT-TO-VIDEO FRAMEWORK

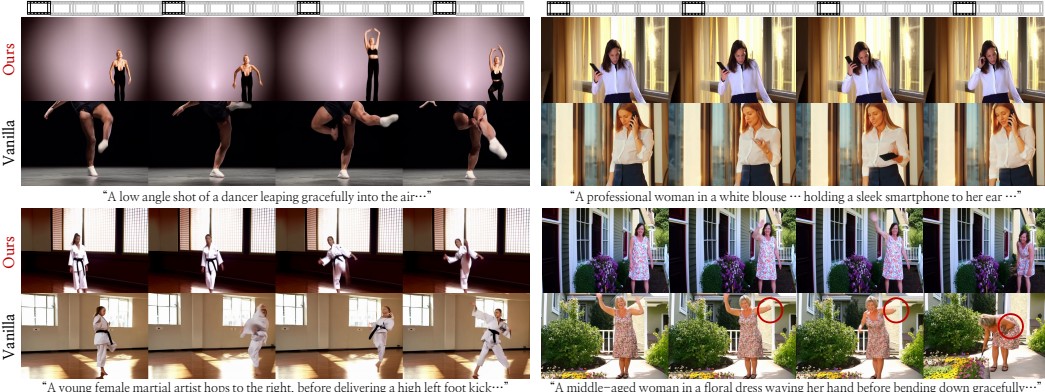

Figure 1: **Qualitative advantage of CAMEO.** Our approach, CAMEO produces more stable and consistent human body articulation in complex motions, whereas vanilla CogVideoX-5B (Yang et al., 2025) often shows pose distortion and inconsistent appearances. For instance, the vanilla model may generate implausible artifacts such as a character repeatedly picking up and putting down a phone. In contrast, our method maintains stable action continuity and prevents such inconsistencies.

## ABSTRACT

Human video generation is becoming an increasingly important task with broad applications in graphics, entertainment, and embodied AI. Despite the rapid progress of video diffusion models (VDMs), their use for general-purpose human video generation remains underexplored, with most works constrained to image-to-video setups or narrow domains like dance videos. In this work, we propose CAMEO, a **CA**scaded framework for general human **M**otion vid**EO** generation. It seamlessly bridges Text-to-Motion (T2M) models and conditional VDMs, mitigating suboptimal factors that may arise in this process across both training and inference through carefully designed components. Specifically, we analyze and prepare both textual prompts and visual conditions to effectively train the VDM, ensuring robust alignment between motion descriptions, conditioning signals, and the generated videos. Furthermore, we introduce a camera-aware conditioning module that connects the two stages, automatically selecting viewpoints aligned with the input text to enhance coherence and reduce manual intervention. We demonstrate the effectiveness of our approach on both the MovieGen benchmark and a newly introduced benchmark tailored to the T2M–VDM combination, while highlighting its versatility across diverse use cases.

## 1 INTRODUCTION

Recent video diffusion models (VDMs) have achieved impressive performance across various video generation tasks, fueled by large-scale datasets and scalable model architectures (Kong et al., 2024;

---

Project page: https://cameo-vdm.github.io/anon4908/

Yang et al., 2025; Wan et al., 2025). Human-centric video generation, as one such task, has emerged as an important avenue of research due to its broad applicability in areas such as digital avatar creation, film production, and fashion (Xu et al., 2025; Kong et al., 2025; Karras et al., 2024). Despite its significance, current VDMs still struggle to handle human-related content, as faithfully capturing articulated body structures and the subtle nuances of human motion remains challenging. Since text alone is often insufficient, yielding unstable results when expressing such detailed structures and dynamics, many works turn to conditional video generative models, which leverage explicit visual cues such as 2D keypoints (Gan et al., 2025), skeletons (Zhu et al., 2024), semantic part masks (Liu et al., 2025b), or geometry-derived signals (Karras et al., 2023; Xu et al., 2024) as conditioning inputs.

While effective, this line of research has predominantly focused on image-to-video (I2V) settings, models trained primarily on narrow-domain datasets such as TikTok (Jafarian & Park, 2021) or UBC fashion videos (Zablotskaia et al., 2019), making them less suitable for generating videos of everyday scenes (Xue et al., 2025). Moreover, these approaches generally overlook the fundamental question of where the motion signals should come from, which is a critical aspect for building a fully end-to-end generative pipeline. Accordingly, T2V approaches have also begun to emerge with the goal of building end-to-end pipelines. One example is HMTV (Kim et al., 2024), which connects a text-to-motion (T2M) model with a 2D skeleton-conditioned video diffusion model to form a T2V pipeline. However, this work merely links off-the-shelf components without specific adaptation for human-centric video generation. For example, the motion produced by the T2M stage is canonical, so the system requires explicit specification of the camera view for rendering, which in this work must be manually provided via text. As a result, the pipeline remains fragmented rather than serving as a complete end-to-end T2V solution.

To overcome these limitations in human-centric video generation, we introduce CAMEO, a **CA**scaded framework for general human **M**otion vid**EO** generation, which couples a T2M model with a conditioned VDM to produce coherent and controllable human motion videos. Our approach consists of two main components: a training strategy that carefully designs textual prompts and conditioning inputs to ensure effective alignment with motion, and an inference-time module that connects the two stages by automatically determining suitable camera views. Together, these components enable an integrated end-to-end pipeline that generates coherent and controllable human videos directly from text, while improving stability and generalization to diverse scenarios. We demonstrate the effectiveness of our approach on the MovieGen benchmark as well as on HuMoBench, a newly introduced benchmark specifically designed for evaluating the integration of T2M and VDM. Beyond evaluation, we further highlight the versatility of our framework through practical use cases, including motion editing and camera view editing.

## 2 RELATED WORKS

### 2.1 CONDITIONED VIDEO DIFFUSION MODELS

As in the image domain, advances in video diffusion models (VDMs) have also led research to increasingly expand toward controllability. A primary direction has been to steer the generation process through conditioning. Conditioning signals have taken various forms, including textual descriptions that specify the content of a video, initial frames that guide subsequent generation, and low-level visual inputs such as edge maps, semantic masks, or depth information (Yang et al., 2025; Wan et al., 2025; Geng et al., 2025; Wang et al., 2024). The latter, in particular, enable structural and contextual control, and have been widely adopted in human-centric video generation, which will be discussed later in this section. A representative framework in this line of work is ControlNet (Zhang et al., 2023), which introduces an additional branch to guide diffusion models with structural signals. Originally developed with UNet backbones, diffusion models for video have now largely transitioned to Diffusion Transformer (DiT)-based architectures (Peebles & Xie, 2023), for which corresponding ControlNet-style conditioning mechanisms have also been developed. For instance, AC3D (Bahmani et al., 2025) demonstrated camera-controllable video generation by applying ControlNet conditioning to CogVideoX (Yang et al., 2025). Motivated by the fact that both camera motion and human motion can be regarded as forms of controllable motion, we design our framework to condition human-related visual cues in a similar manner.

## 2.2 Human Motion Video Generation with Diffusion Models

Methods for human video generation can be broadly categorized into two primary approaches: vision-driven and text-driven. Vision-driven approaches rely on explicit visual cues, which are then used to guide the generation process. These cues often take the form of 2D signals such as keypoints (Gan et al., 2025), skeletons (Wang et al., 2025), and semantic masks that capture body articulation (Liu et al., 2025b). They can also include 3D or geometric information such as depth maps, normal maps, or dense correspondence maps or even renderings of parametric 3D models (Karras et al., 2023; Xu et al., 2024; Zhu et al., 2024; Cao et al., 2025). However, this reliance on detailed visual inputs, while enabling precise control, has largely confined the application of these image-to-video (I2V) methods to narrow domains. Moreover, they remain restricted to settings where motion cues are extracted from external videos, with little exploration of more flexible approaches to obtain such signals directly. Beyond these I2V settings, text-driven approaches have been relatively less explored, with only a few notable examples. For instance, Text2Performer (Jiang et al., 2023) generates videos that follow the intended action described in text. However, it also requires a reference image, it cannot be regarded as a fully text-to-video approach. More recently, HMTV (Kim et al., 2024) further extended this direction by explicitly coupling text-to-motion (T2M) generation with motion-conditioned video synthesis, demonstrating the potential of text-driven pipelines for controllable human video generation. To further unlock this potential, we propose methods that enhance the generative ability of each stage and, more importantly, introduce dedicated modules that enable a fully integrated text-to-video (T2V) pipeline.

## 2.3 Text-to-Motion Generation with Diffusion Models

Human motion can be represented in multiple ways, such as joint positions, joint rotations, or parametric models like SMPL (Loper et al., 2023). Building on these representations, T2M generation has been extensively studied as a means to bridge natural language and 3D human motion. In particular, recent advances in diffusion models have propelled this direction, enabling substantial progress in synthesizing realistic and semantically aligned motions from text (Tevet et al., 2023; Yuan et al., 2023; Zhang et al., 2024). These models provide an intuitive way to generate human movements directly from language, offering flexible control of animated characters with applications in VR, gaming, HCI, and robotics. However, most studies have treated T2M as an isolated task, with relatively few extending it to broader video generation pipelines. A notable exception is Move-in-2D (Huang et al., 2025), which conditions motion on 2D inputs and demonstrates an application to video synthesis, though it does not establish a full T2V pipeline. To better leverage these advances, we establish a complete T2V pipeline by integrating T2M models with video diffusion models, maximizing their complementary strengths. For the T2M component, we adopt STMC (Petrovich et al., 2024) for its strong capability in fine-grained temporal and compositional control, making it well-suited for generating coherent and continuous motions in video generation.

## 3 Method

Our key idea is to build a unified framework by integrating Text-to-Motion (T2M) models and Video Diffusion Models (VDMs), two powerful approaches whose combined potential remains largely unexplored. We realize this idea through CAMEO, a cascaded yet integrated design for human-centric text-to-video generation. The method consists of two main components: a training strategy and inference procedure. In Sec. 3.1, we first describe how to construct and adapt visual and textual conditions to effectively guide our VDM in the human motion domain. In Sec. 3.2, we present our inference procedure, which employs stage-specific prompting and a camera view selection module that bridges the two models. An overview of our framework is provided in Fig. 2.

## 3.1 Data Conditioning Strategy for Training VDM

Our primary objective is to train a VDM conditioned on both a visual motion signal and a textual description. This goal necessitates a training dataset composed of triplets, $(x, m, t)$, containing the source video $x$, the visual motion condition $m$, and the text condition $t$, respectively. For our training data, we chose HOIGen-1M (Liu et al., 2025a) and Motion-X++ (Zhang et al., 2025), leveraging the former's coverage of daily-life activities and the latter's diverse, complex motions. However, these

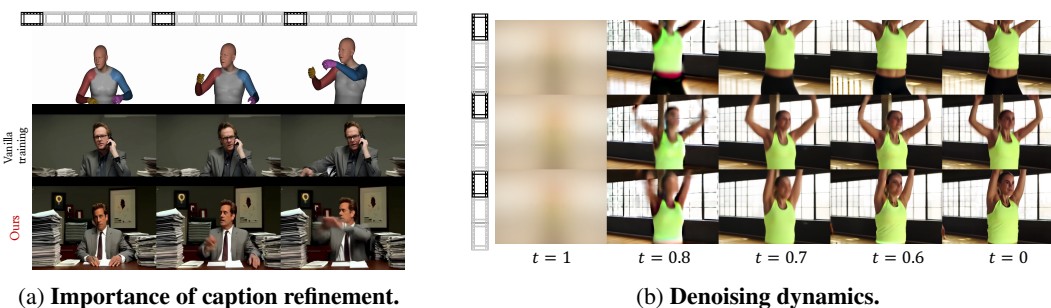

Figure 2: **Overview of CAMEO.** Given a text prompt, we first disentangle it to separate motion-related and semantic components. The motion prompt is converted into an initial motion sequence via a text-to-motion model. The sequence is rendered as SMPL-based guidance videos, where a camera-aware conditioning module determines the viewpoints for rendering. Finally, the video diffusion model synthesizes the human video, guided by the semantic prompt and motion condition, seamlessly bridging text-to-motion and text-to-video generation.

(a) **Importance of caption refinement.**          (b) **Denoising dynamics.**

Figure 3: **Analysis of training choices.** (a) The model trained with the original coarse captions often fails to learn fine-grained motion details, whereas the model trained with our refined captions converges faster and produces more accurate motion details. (b) Macro body movements emerge early in the denoising process, while finer details such as clearer body outlines appear later, around $t = 0.6$.

datasets are not fully aligned with our needs: HOIGen-1M does not provide SMPL annotations, and the captions in both datasets have limitation for our purpose. To address this, we design a dedicated pipeline to construct them.

**Refining text prompt**    In particular, the textual prompts required a separate treatment. The captions provided in the datasets often mixed descriptions of successive motions with scene or appearance details. Our initial attempts on training the conditional VDM naively with the captions provided in the datasets showed conflict between the motion information against the visual conditions, consequently hindering effective training. This is illustrated in Fig. 3a, which presents early-stage inference results during training: the top row (original captions) shows a model conditioned on coarse character location but failing to capture fine-grained motions, while the bottom row (refined captions) shows faster convergence with more accurate motion details. Furthermore, Motion-X++ contains videos recorded in laboratory environments, yet its captions omit such context and provide only the sparse semantic descriptions. As a result, a model trained on such data often relied on spurious correlations, producing videos that reflected lab-like settings rather than diverse environments.

To address these issues, we first recaptioned Motion-X++ using a vision-language model (VLM) to enrich its semantic descriptions and contextual details omitted in the original captions. Building on this, we applied a large language model (LLM) across all datasets to restructure the textual annotations $t$ into two complementary parts: motion caption $t_m$ and semantic caption $t_s$, which

capture contextual and scene-level information. We then utilized $t_s$ to train the VDM, while $t_m$ was employed to guide the text-to-motion model during generation. See Appendix B for details.

**Preparing visual motion cues**  We constructed the visual motion condition $m$ by rendering 3D SMPL meshes.[1] For HOIGen-1M, which lacks SMPL annotations, we estimate per-frame SMPL parameters from the videos $x^{1:N}$ using SMPLest-X (Yin et al., 2025); for Motion-X++, we directly use the provided annotations. The resulting meshes are then rendered with lighting from their corresponding estimated camera poses and assigned distinct colors to body-part regions (e.g., head, torso, and limbs). This produces $m^{1:N}$ that encode both global body layout and localized articulation, making them a more informative signal for motion-conditioned video generation.

**Training conditioned VDM**  To enable conditioning, we adopt a ControlNet-based conditioning approach (Zhang et al., 2023). In particular, we adapt the architecture of AC3D (Bahmani et al., 2025), which shares our objective of achieving precise motion control and demonstrated strong results. At the same time, human motion video generation poses requirements beyond generic architectures: while camera control can often be achieved at a coarse level, human motion demands both large-scale motion control and fine-grained articulation of body parts. Accordingly, we design tailored conditioning strategies that leverage SMPL-based cues to support precise motion guidance and highlight structural outlines along the denoising trajectory. This is evident in Fig. 3b, where large-scale body movements are established very early in the denoising process, whereas finer details such as clearer body outlines are not fully produced until around $t = 0.6$. Furthermore, we observe larger variability in the timesteps required to achieve proper conditioning. Motivated by these observations, when sampling the diffusion timestep during training, we adopt a truncated normal distribution over the range $[0.6, 1]$ as in AC3D, but 1) reduce the mean from 0.95 to 0.9 to account for the fine-grained body outlines that appear later in the timestep, and 2) increase the standard deviation from 0.1 to 0.2 to account for the larger variability. During inference, the conditioning is applied only within this range of timesteps. Once the training is complete, we proceed to the inference stage, where T2M generation is followed by VDM synthesis.

### 3.2 END-TO-END INFERENCE: FROM T2M TO VDM

**Stage1: Text-to-Motion generation**  The first stage of our inference process is to generate a motion sequence handled by a T2M model $\mathcal{M}$. This model is conditioned on the motion prompt $t_m$, obtained from our prompt disentanglement stage, which allows T2M to focus on motion-specific information and thereby generate more accurate motions. It then outputs a sequence of low-dimensional SMPL parameters, which are further converted into human body meshes represented by their 3D vertices $V_{1:K}$ via the SMPL model.[2] This process is defined as:

$$V_{1:K} = \mathcal{M}(t_m), \quad V_i \in \mathbb{R}^{N_v \times 3} \tag{1}$$

where $K$ is the number of frames, $N_v$ is the number of mesh vertices (e.g., 6890 for SMPL), and $V_i$ contains the 3D coordinates of $(x, y, z)$ of all vertices at frame $i$. After obtaining the sequence of 3D vertices, we project them into 2D space to be used as conditioning signals. (See Fig. 2, 1) for an illustration.)

**Stage2: Camera viewpoint selection & Conditioned video generation**  A crucial step here is determining the camera parameters, since they define how the 3D meshes are viewed in the 2D plane, thereby shaping the composition of the scene, including the subject's location in the frame and its apparent scale. To address this, we propose a text-aware camera selection module that automatically determines suitable viewpoints for each generated video. Given that off-the-shelf models for text–camera alignment are not available, and that training a dedicated module solely for this purpose would be impractical, we exploit the generative prior of video diffusion models, which implicitly captures $p(\text{camera} \mid \text{text})$ within their training objective of approximating $p(\text{video} \mid \text{text})$.

Concretely, we first use the original prompt $t$ from the dataset-which contains both motion and semantic information-to generate a reference video with the video diffusion model. We stop generation

---

[1]Here and throughout the paper, we use SMPL for simplicity, although our pipeline is compatible with both SMPL and SMPL-X.

[2]When instantiated with SMPL instead of SMPL-X, the face and the hand is set as their default template form.

Table 1: **Quantitative results for each benchmark.** Our method achieves top performance across benchmarks, excelling in both motion consistency and semantic alignment. Vanilla models without visual conditioning show weak consistency, while other baselines that share the same motion conditions as ours achieve reasonable consistency but fall behind in image quality and text alignment. **Bold**: Best, Underline: Second Best.

| Benchmark | Method | Appearance Metrics | | | | Motion Metrics | | Prompt Fidelity |
|---|---|---|---|---|---|---|---|---|
| | | Aesthetic Quality | Image Quality | Subject Consistency | Background Consistency | Motion Smoothness | Dynamic Degree | Text Alignment |
| MovieGen -Human Activity (Polyak et al., 2024) | Vanilla | 0.539 | 0.592 | 0.933 | 0.950 | 0.981 | **0.644** | **0.272** |
| | HMTV | 0.485 | **0.620** | 0.946 | 0.945 | **0.987** | 0.570 | 0.254 |
| | CamAnimate | 0.441 | 0.580 | **0.961** | **0.958** | 0.981 | 0.585 | 0.226 |
| | Ours | **0.548** | 0.613 | 0.952 | 0.957 | 0.983 | 0.545 | 0.258 |
| HuMoBench | Vanilla | **0.538** | 0.587 | 0.918 | 0.943 | 0.981 | 0.608 | 0.250 |
| | HMTV | 0.526 | 0.614 | 0.949 | 0.947 | 0.989 | 0.641 | 0.252 |
| | CamAnimate | 0.441 | 0.601 | 0.954 | 0.951 | 0.989 | **0.775** | 0.248 |
| | Ours | 0.535 | **0.615** | **0.955** | **0.957** | **0.990** | 0.567 | **0.262** |

at an early denoising stage where the human shape becomes is sufficiently discernible for camera extraction and use this partial output as an approximation. This procedure also keeps the computation overhead minimal. We then compute the camera parameters $(R_i, T_i)$ from the reference video that best aligns the extracted vertices with the generated frames. Assuming that the vertices generated by our T2M model, $V_{1:K}$, lie in the same canonical space, we can render them into the reference view using the estimated camera parameters:

$$\boldsymbol{m}_i = \Pi(R_i V_i^{\top} + T_i), \quad \boldsymbol{m}_i \in \mathbb{R}^{N_v \times 2}, \tag{2}$$

where $\Pi(\cdot)$ denotes the perspective projection with estimated camera intrinsics, and $\boldsymbol{m}_i$ gives the 2D coordinates of the projected vertices at frame $i$, as shown in Fig. 2, 2). This ensures that the human generated by the T2M model is placed in the corresponding location of the reference scene. Conditioned on the semantic prompt $t_{\text{sem}}$ and the visual motion signal $\boldsymbol{m}$, we then perform inference with the trained VDM (See Fig. 2, 3)). Consequently, CAMEO generates videos with camera views naturally aligned to the given text, without requiring manual effort for camera view selection.

## 4 EXPERIMENTS

### 4.1 IMPLEMENTATION DETAILS

In Stage 1, we employ the off-the-shelf STMC (Petrovich et al., 2024), a text-to-motion (T2M) approach that directly outputs SMPL pose parameters to produce 6-second motion sequences at 23fps. In Stage 2, we build on top of the CogVideoX-5B text-to-video diffusion model (Yang et al., 2025), augmenting it with a ControlNet-style branch attached to the first 21 transformer layers. For training, we use AdamW with $\beta_1 = 0.9$, $\beta_2 = 0.95$, a cosine learning rate schedule with a base learning rate of $1 \times 10^{-4}$, and a batch size of 16 with gradient accumulation. At inference, we apply a guidance scale of 4.0 and a pose guidance scale of 2.0, generating 6-second videos with 49 frames at 8 fps. The input motions from Stage 1 are downsampled from their native 23 fps to align with this frame rate. For Motion-X++ caption enrichment, we employed InternVL2.5-38B (Chen et al., 2024) as the VLM to augment semantic descriptions and contextual details. Beyond this dataset-specific augmentation, we further adopted Qwen3-32B (Qwen, 2025) as the LLM for caption refinement and input formatting across all datasets. Detailed prompting strategies are provided in the Appendix B. All experiments, including fine-tuning and inference, are conducted on 40GB A6000 GPUs.

### 4.2 QUANTITATIVE EVALUATION

**Benchmarks** We use two benchmarks for evaluation. First, we consider 204 prompts from the human-activity category of the MovieGen (MGen) benchmark (Polyak et al., 2024). Additionally, we introduce HuMoBench, a benchmark for evaluating pipelines that link T2M models with motion-conditioned video generation, addressing the lack of an existing protocol. HuMoBench contains 120 entries, each consisting of three components: (1) a plain-text motion prompt, (2) a plain-text semantic prompt describing a plausible context, and (3) a combined prompt merging motion and semantic information. To construct the entries, we first employed an LLM to generate 120 motion

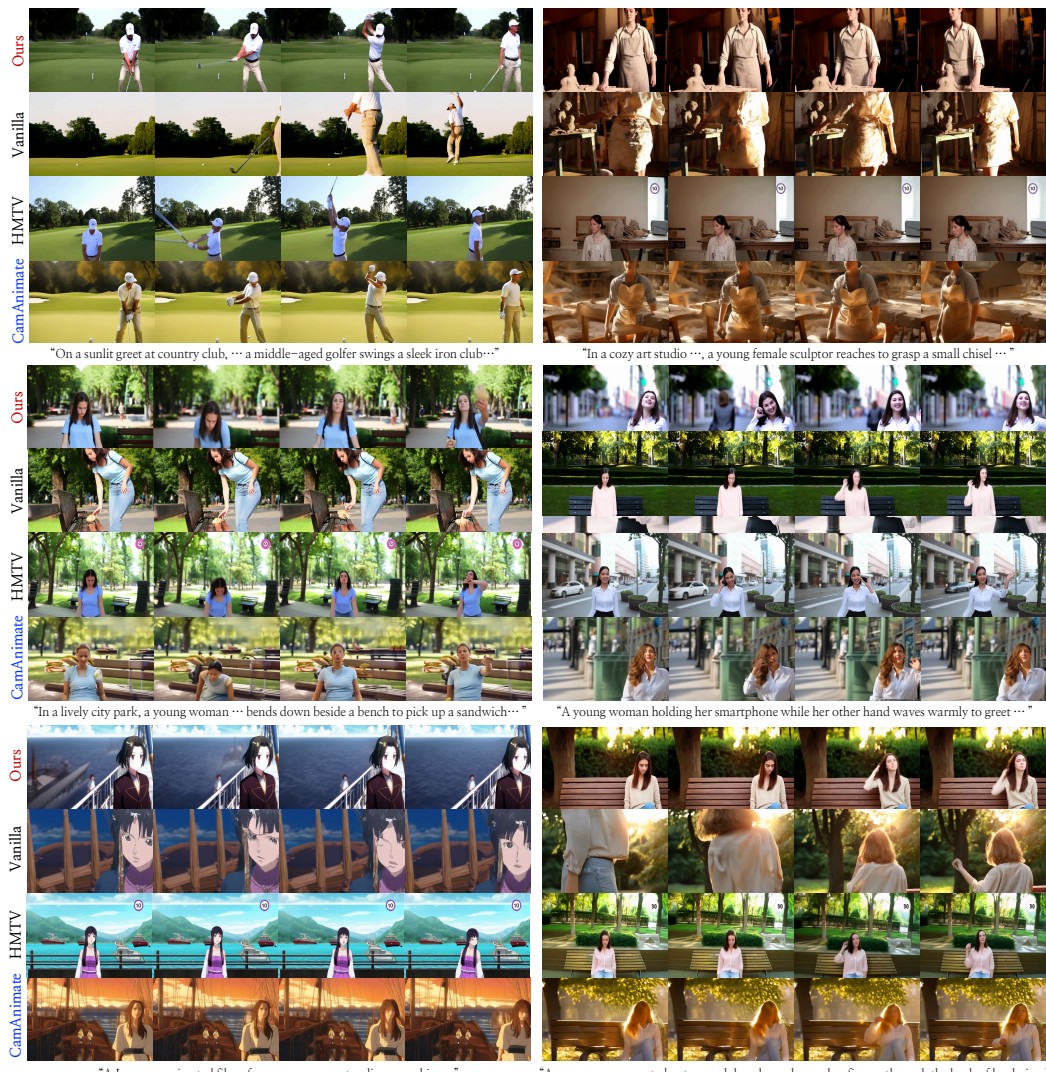

Figure 4: **Qualitative comparisons.** Our pipeline captures complex human structures and motions more faithfully than baseline models, while also producing more natural and consistent camera views.

descriptions at the granularity understood by T2M models. Subsequently, the LLM was instructed to infer plausible scenarios for each sequence and, on this basis, generate the remaining prompts. The exact prompts and several representative benchmark examples are provided in Appendix B.

**Baselines** We benchmark our models against their base (pre-trained) versions, as well as HMTV (Kim et al., 2024) and CamAnimate (Wang et al., 2024). HMTV was originally defined with a unified prompt guiding a motion generator, which in turn provides keypoint conditioning to a video model. However, as the benchmark tasks involve more diverse and challenging motions, directly generating motion from the given prompt without additional strategy becomes infeasible. To enable a fair comparison, we retain our motion generation pipeline; however, because HMTV requires the camera view to be manually specified—a detail not typically available in the prompt—we place the camera at the center of the scene. We then train the video diffusion model with a conditioning scheme analogous to HMTV's keypoint-based design, without our additional strategies such as text refinement and timestep control. We also include CamAnimate (Wang et al., 2024), an image-to-video (I2V) baseline for human animation trained on HumanVid. Since strong text-to-video models tailored to our domain are not yet available, we adapt CamAnimate within our framework. Specifically, we condition on keypoints following its original design, generating an initial frame from a skeleton input via text-to-image synthesis and then applying the I2V model to produce the remaining frames.

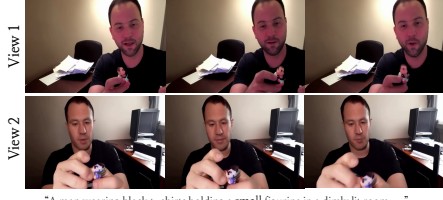

| (a) **Motion editing results.** | (b) **Camera view editing results.** |

Figure 5: **Extensions.** (a) Motion editing: Initial motions are regenerated with SDEdit (Meng et al., 2022), yielding variations while preserving overall semantics. (b) Camera view editing: guidance videos are rendered with perturbed extrinsics, producing different viewpoints while preserving semantics.

**Metric - VBench**   We follow the quantitative evaluation protocol of Chefer et al. (2025), which uses VBench (Huang et al., 2024) to evaluate video generators. While Chefer et al. (2025) reports only appearance and motion scores, we additionally report the VBench text alignment score which is based on CLIP Similarity. As shown in Tab. 1, our method achieves the best or second-best performance across nearly all metrics. This demonstrates its balanced strength in both appearance and motion quality, as well as superior text alignment. In particular, the largest margin over the vanilla model appears in the consistency metrics. As also illustrated in the qualitative examples, vanilla models often fail to maintain stable articulation under complex motions, leading to entangled or distorted limbs. By contrast, our method exhibits far fewer such failures, which likely accounts for the improved consistency scores. Other baseline models that leverage our T2M results also achieve reasonable performance on consistency and motion smoothness, since the underlying motions are shared. However, they fall short in aesthetic and image quality, largely due to the absence of a camera-view module and the weaker VDM used in CamAnimate.

### 4.3 QUALITATIVE EVALUATION

Comprehensive qualitative results are shown in Fig. 4. The vanilla model often fails to reproduce systematic motions, such as a golf swing, and struggles with basic actions by either stalling or exhibiting unnatural velocities. While motion-conditioned methods yield more coherent results, they introduce other challenges. CamAnimate, for instance, struggles with semantic fidelity and often introduces noticeable visual artifacts. Similarly, HMTV represents motion reasonably well but lacks a text refinement strategy. Consequently, it fails to disentangle textual artifacts from the motion data, unnecessarily rendering elements like video subtitles from the training dataset. Our model addresses this by utilizing refined captions that provide richer semantic descriptions. Furthermore, HMTV's absence of a camera-view selection mechanism leads to monotonous and often suboptimal viewpoints. In contrast, our method produces diverse camera perspectives, from upper-body to full-body shots, that are semantically aligned with the text to best depict the scene.

### 4.4 USER STUDY

We conducted an A/B study by showing participants paired videos from a baseline and our method for the same prompts. The baselines matched those in the qualitative comparison (Vanilla, HMTV, CamAnimate), with three samples each, yielding nine pairs. Participants judged Motion / Action Quality and Video / Visual Quality, choosing win, lose, or tie. Detailed participant instructions are provided in the Tab. 11. Among 44 participants, our method was preferred overall(Tab. 2).

|  | **Win** | **Lose** | **Tie** |
|---|---|---|---|
| **Motion Quality** | **0.631** | 0.214 | 0.145 |
| **Visual Quality** | **0.545** | 0.257 | 0.186 |

Table 2: **User study results for motion and visual quality.** Win indicates the proportion of cases where Ours was preferred over the baseline.

### 4.5 ABLATION STUDIES

**Importance of text refinement strategy**   To assess the effect of our text refinement strategy, we train and evaluate the VDM using the original dataset captions, which mix both motion and semantic information. In the case of Motion-X, these captions omit elements such as lab environments or subtitles

that can significantly influence video quality, making disentanglement more challenging. All other settings and sampling are kept identical to our main model. On HuMoBench, text refinement yields consistent gains on most metrics, while the non-refined variant attains a higher motion score, likely due to the metric's bias toward dynamics. Considering the full set of metrics, our method achieves improvements without compromising overall consistency. Beyond metrics, HMTV results in Fig. 4 show that models without this strategy fail to disentangle dataset artifacts and often reproduce them in generated videos.

**Importance of view selection module** We also ablate the view selection module by fixing the camera at the scene center. As shown in Fig. 3 and Fig. 6, this leads to monotonous, suboptimal viewpoints and a slight drop in quality. We conjecture that this occurs because VDMs perform better when inputs are closer to their training distribution; thus, adaptive view selection provides more in-domain views and ultimately improves performance. We report results averaged over HuMoBench in the main text, with detailed results for both ablation variants provided in Tab. 6 of the Appendix.

### 4.6 Extensions

In addition to evaluation, we demonstrate extensions that are uniquely enabled by our cascaded approach. In particular, we demonstrate motion editing and camera view editing as representative use cases, highlighting the broader applicability of our approach.

| Model | Appearance Metrics | Motion Metric | Prompt Fidelity |
|---|---|---|---|
| Ours | 0.768 | 0.779 | 0.262 |
| w/o Refine | 0.761 | 0.791 | 0.261 |
| w/o View Module | 0.751 | 0.774 | 0.261 |

Table 3: **Quantitative results for ablation studies.** *w/o Refine*: ablated on text refinement; *w/o View Module*: ablated on view selection.

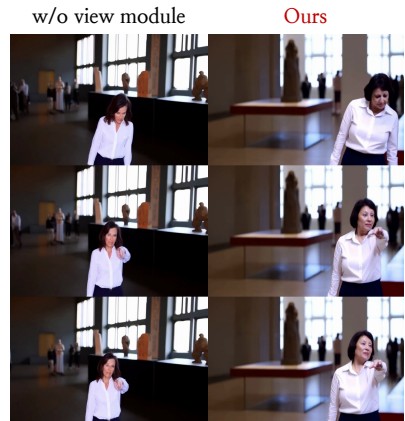

w/o view module          Ours

"A museum curator, turns her head to the left and extends her arm to point⋯ "

Figure 6: **Qualitative results: ablation on view selection.** View selection leads to improved quality.

**Motion editing** For motion editing, we first generate an initial motion sequence from the text prompt. We then apply the SDEdit (Meng et al., 2022) technique by re-noising the sequence to an intermediate diffusion step and regenerating it with the same text input. By conditioning these edited motions in our staged pipeline, users can refine motion details and obtain videos that better align with their intended outcomes, as illustrated in Fig. 5a.

**Camera view editing** Another extension is to keep the motion fixed while modifying the camera viewpoint. We perturb the estimated camera extrinsics by small rotations or translations and render new guidance videos accordingly. Conditioning the video diffusion model on these variants produces human videos from slightly different perspectives while largely preserving semantic consistency, as illustrated in Fig. 5b.

## 5 Conclusion

In this work, we present CAMEO, a text-to-video pipeline that is both cascaded and integrated, seamlessly bridging text-to-motion (T2M) and video diffusion models (VDM). To complete the pipeline, we propose dedicated strategies and modules for human motion generation. First, to enable effective training of the conditioned VDM, we redesign textual captions to mitigate conflicts between the text-to-motion and video diffusion stages, and adopt training configurations better suited for human motion generation. At the inference stage, we introduce a camera-view selection module that encourages both natural perspectives and consistent appearance across frames. These contributions allow our method to faithfully capture complex human structures and motions in natural scenes. Through extensive experiments, we demonstrate the effectiveness of our approach in both quantitative and qualitative evaluations, as well as several practical use cases.

**Limitations** Our approach relies on an off-the-shelf T2M model as the starting point, which inherently binds the performance of the overall pipeline to that of the underlying T2M model. For example, current models still struggle to capture fine-grained finger articulation, leading to limitations in generating highly detailed motions. Moreover, since our training is based on datasets with primarily single-person scenes, the model may generalize poorly to out-of-domain scenarios such as crowded or multi-person environments. These limitations are likely to diminish as both T2M model and VDM advance, enabling finer-grained motion capture and stronger generalization to diverse and complex scenes.

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

# A FURTHER EXPERIMENTS

In this section, we present additional experiment results, including qualitative visualizations of generated videos and further ablation studies.

## A.1 QUALITATIVE RESULTS

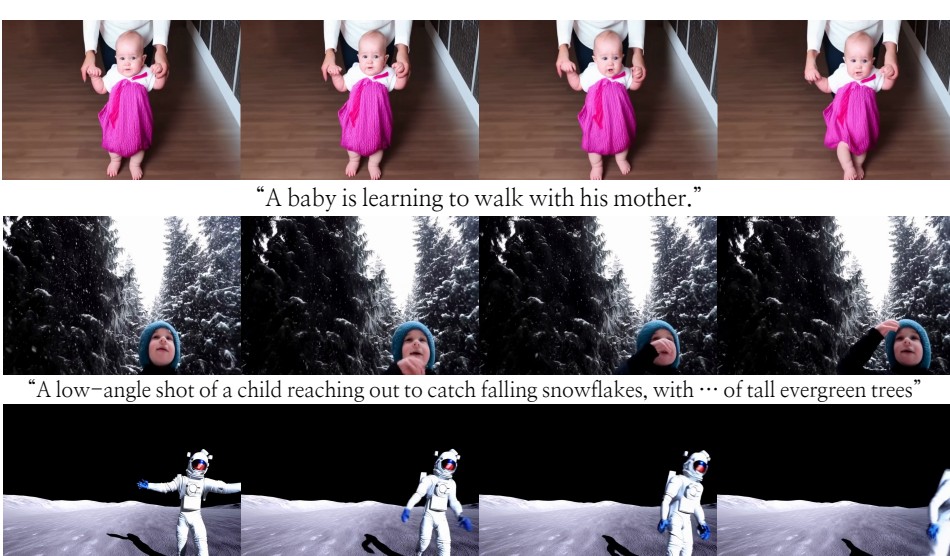

"A baby is learning to walk with his mother."

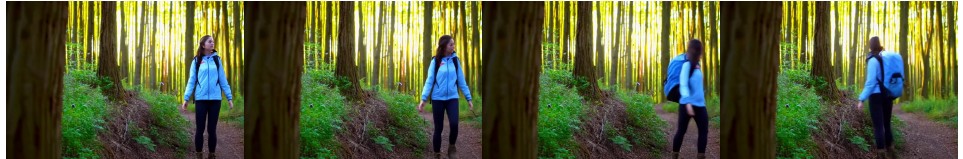

"A low−angle shot of a child reaching out to catch falling snowflakes, with ⋯ of tall evergreen trees"

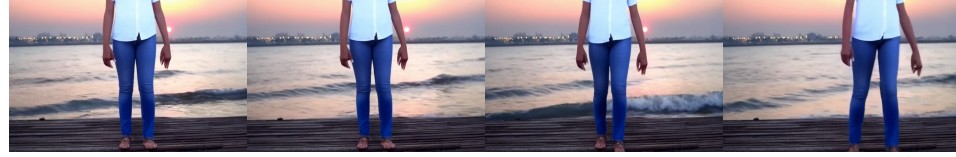

"An astronaut runs on the surface of the moon, the low angle shot shows the vast background of the moon⋯"

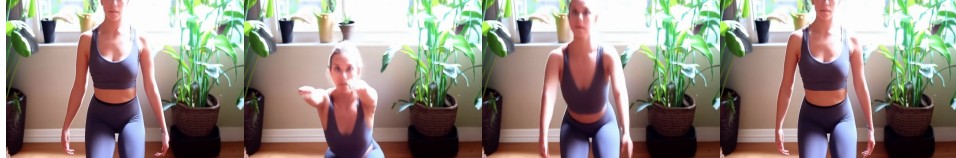

"A young woman in hiking gear walks steadily along a sunlit forest trail ⋯ "

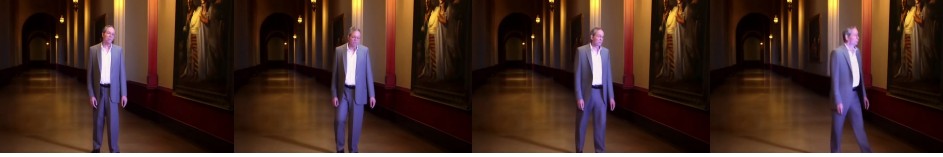

"a woman wearing blue jeans and a white t shirt taking a pleasant stroll in Mumbai Indiaduring a beautiful sunset"

"In a home gym adorned with green plants⋯, a young woman bends her knees to lower into a deep squat, "

"A middle−aged man in a tailored gray suit slowly walks down a dimly lit hallway of a museum⋯ "

Figure 7: **Additional qualitative results.** Representative video samples generated by our method, illustrating its ability to handle diverse actions and scenes with stable articulation and consistent appearances.

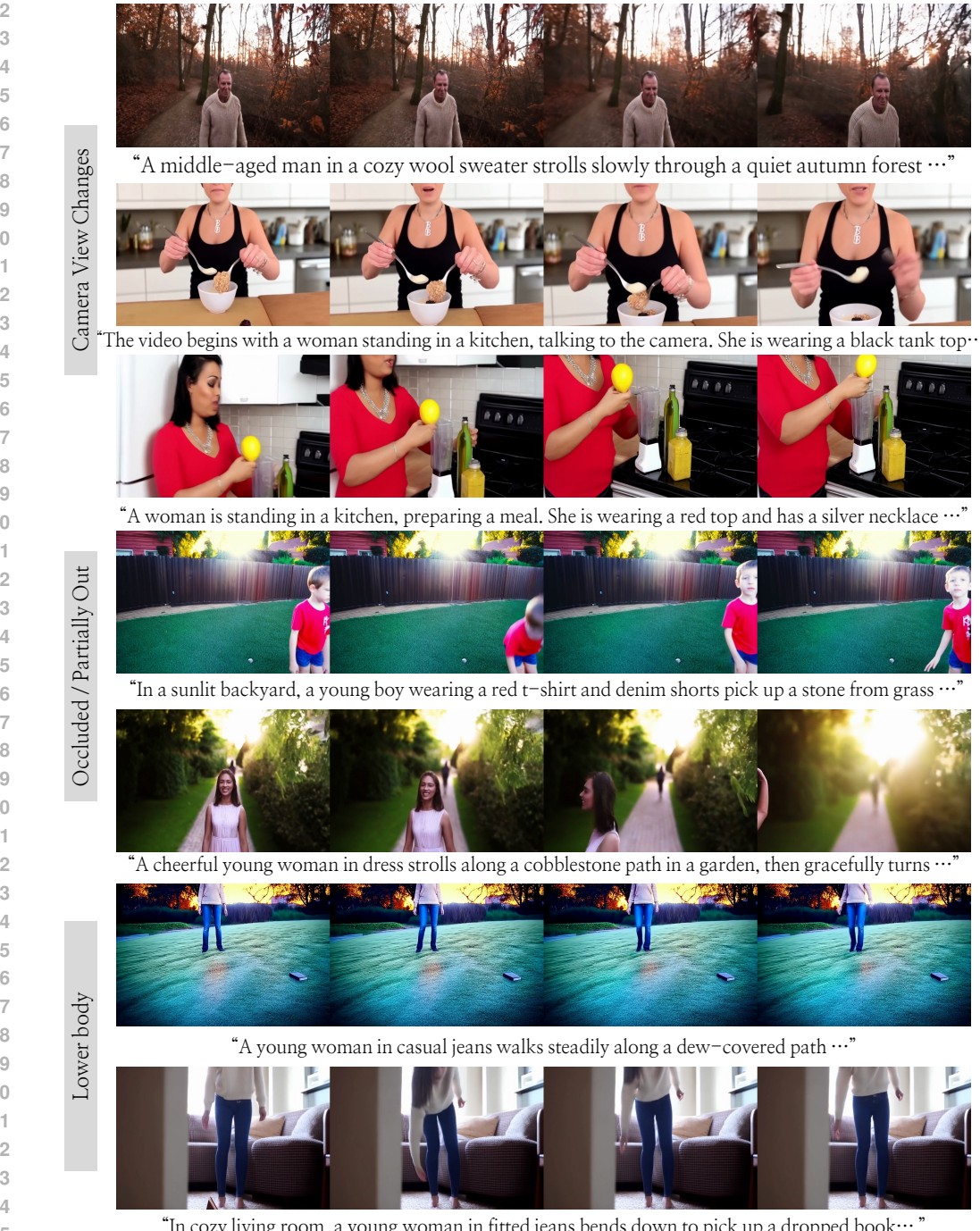

Figure 8: **Additional qualitative results.** Representative video samples generated by our method, illustrating its ability to generalize across challenging scenarios such as rapidly changing camera views, partially visible subjects, and varied body-region visibility, while maintaining stable articulation and consistent appearances.

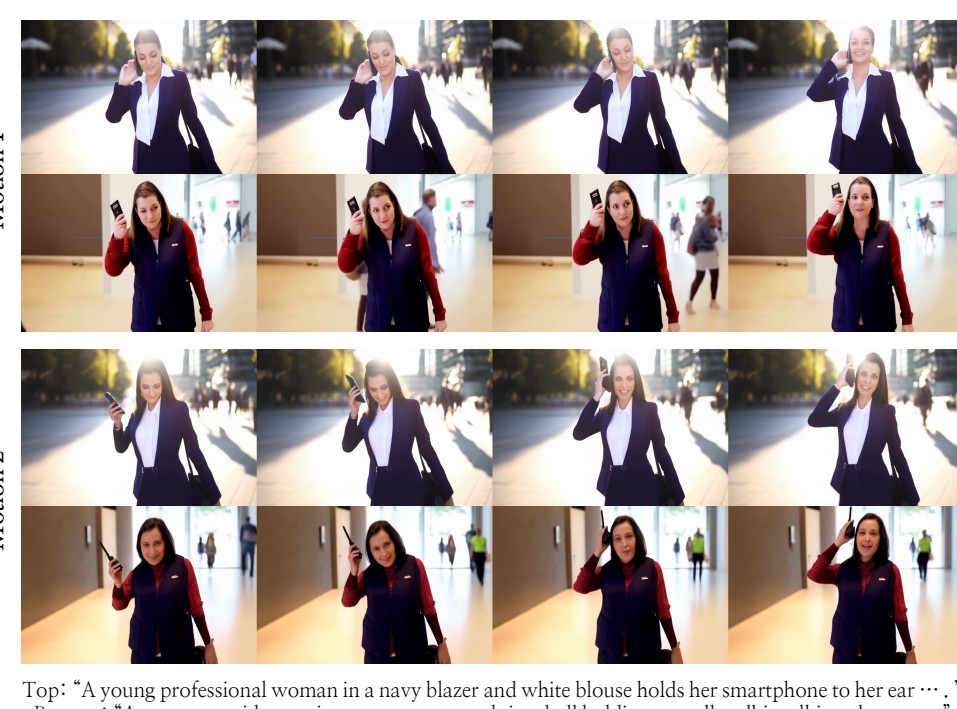

Top："A young professional woman in a navy blazer and white blouse holds her smartphone to her ear ⋯ ．"
Bottom："A museum guide wearing a navy vest stands in a hall holding a small walkie talkie to her ear ⋯"

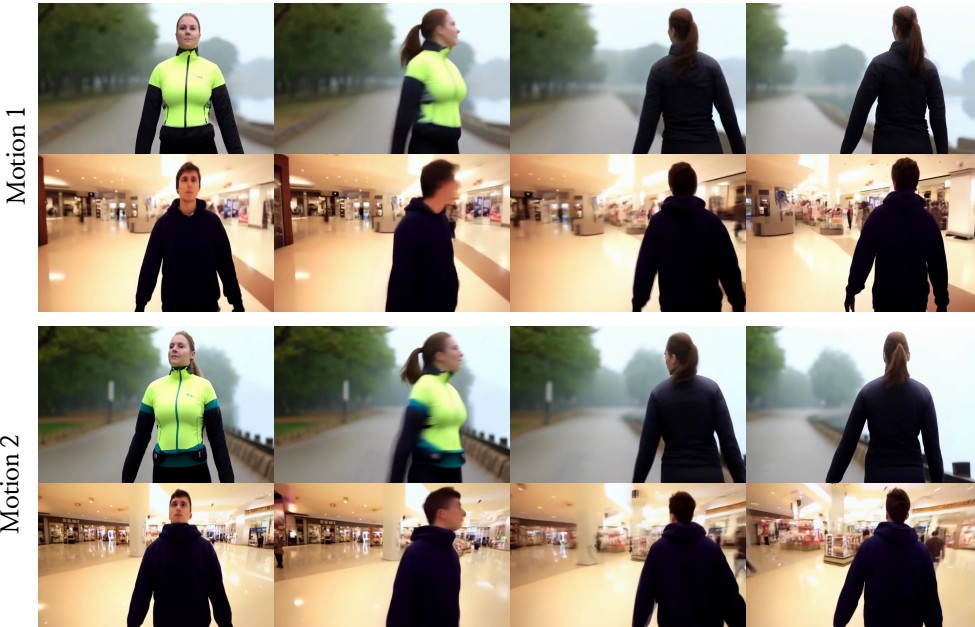

Top："In the morning mist of a quiet city park, a young woman in neon running jacket smoothly pivots to the right⋯ "
Bottom："Young man walking through an indoor mall, wearing a navy hoodie, looking around calmly⋯"

Figure 9: **Additional motion editing results.** Supplementary examples of motion editing.

### A.2 EXTENSION - MOTION EDITING

### A.3 ADDITIONAL ABLATION STUDIES

**Full metrics for ablation studies** Tab. 4 reports the full set of evaluation metrics corresponding to Fig. 3 in the main paper. Without the text refinement strategy, metrics such as consistency and motion smoothness degrade. This suggests that, when both semantic and motion information are mixed in

Table 4: **Full quantitative results for ablation studies.** We report detailed per-benchmark results for the ablations on text refinement and the view selection module. *w/o Refine* denotes the variant without text refinement, and *w/o View Module* denotes the variant without the view selection module.

| Method | Appearance Metrics | | | | Motion Metrics | | Prompt Fidelity |
|---|---|---|---|---|---|---|---|
| | Aesthetic Quality | Image Quality | Subject Consistency | Background Consistency | Motion Smoothness | Dynamic Degree | Text Alignment |
| Ours | 0.535 | 0.615 | 0.955 | 0.957 | 0.990 | 0.567 | 0.262 |
| w/o Refine | 0.540 | 0.613 | 0.936 | 0.956 | 0.982 | 0.600 | 0.261 |
| w/o View Module | 0.528 | 0.585 | 0.944 | 0.946 | 0.989 | 0.558 | 0.261 |

Table 5: **Additional ablation studies: benefit of view selection over additional inference steps and conditioning layer depth.** Among the tested configurations, the 21-layer conditioning achieved the best overall performance on HuMoBench.

| Method | Appearance Metrics | | | | Motion Metrics | | Prompt Fidelity |
|---|---|---|---|---|---|---|---|
| | Aesthetic Quality | Image Quality | Subject Consistency | Background Consistency | Motion Smoothness | Dynamic Degree | Text Alignment |
| Ours (21 layers) | 0.535 | 0.615 | 0.955 | 0.957 | 0.990 | 0.567 | 0.262 |
| w/o View Module + longer inference | 0.526 | 0.576 | 0.946 | 0.949 | 0.991 | 0.567 | 0.263 |
| 42 layers | 0.528 | 0.585 | 0.944 | 0.946 | 0.989 | 0.558 | 0.261 |
| 10 layers | 0.531 | 0.624 | 0.951 | 0.962 | 0.986 | 0.471 | 0.258 |

the captions, the VDM conditioning becomes noisier, leading to less stable and coherent motions. In contrast, removing the view selection module primarily leads to drops in aesthetic quality and image fidelity, highlighting its role in producing visually appealing and coherent frames.

**Comparison between view selection and additional inference steps** Our view selection module introduces computational overhead. Therefore, a critical question is whether this additional computational budget could be more effectively utilized by the baseline model, for instance, by increasing its number of diffusion steps. To address this, we compare our method against a baseline variant whose inference time is extended to match the computational cost of our module. Specifically, since our method performs view selection using videos generated with about 15 denoising steps, we account for this overhead and compare against a baseline that generates videos with 70 denoising steps. The results show that while some metrics exhibit marginal improvements, the gains are limited.

**Number of conditioned layers** The number of layers to which ControlNet conditioning is applied directly affects training efficiency, memory consumption, and overall resource usage. To identify an effective yet efficient configuration, we varied the number of conditioned layers and trained models under each setting. We then evaluated their performance on our benchmark and selected the final configuration based on both accuracy and stability. As shown in Tab. 5, conditioning 21 layers provided the best trade-off and yielded the strongest overall performance.

## A.4 FURTHER ANALYSIS

**Analysis of Camera Viewpoint Distribution** To further assess the diversity of camera viewpoints in our system, we analyzed the distribution of camera translation vectors obtained from the reference videos that guide view selection. For the 120 HuMoBenchclips, each reference video provides a single translation vector of dimension (3) extracted from its first frame. We visualized these vectors using a scatter plot and summarized their statistics. The distribution exhibits a wide spread along all three axes, demonstrating that the generated reference views span a broad range of camera positions rather than collapsing to a narrow region. This coverage supports the view selection module by offering varied positional cues during generation.

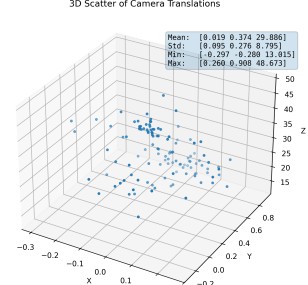

Figure 10: **Camera Viewpoint Diversity.** Distribution of extracted 3D camera translation vectors from 120 HuMoBench reference videos.

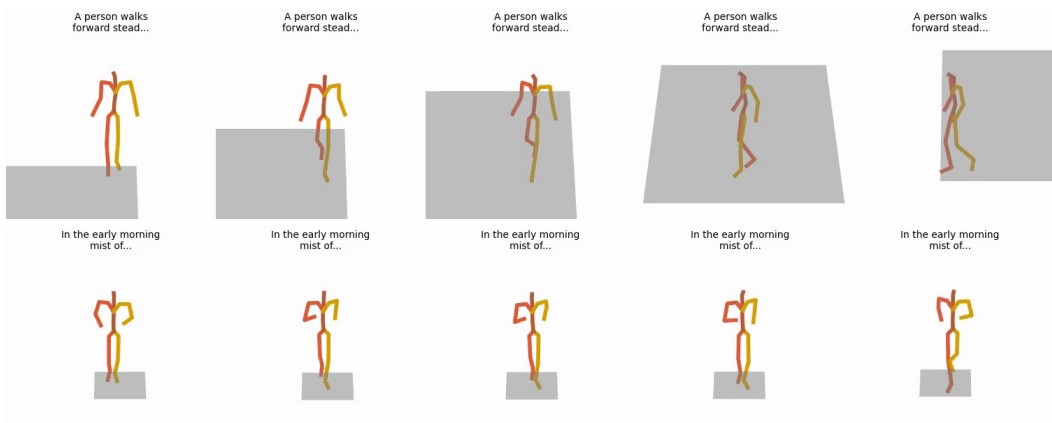

Top: A person walks forward steadily, then pivots their body by turning to the right
Bottom: In the early morning mist of a quiet city park, a young woman ⋯ jogs steadily,
⋯ then smoothly pivots to the right to follow the trail⋯

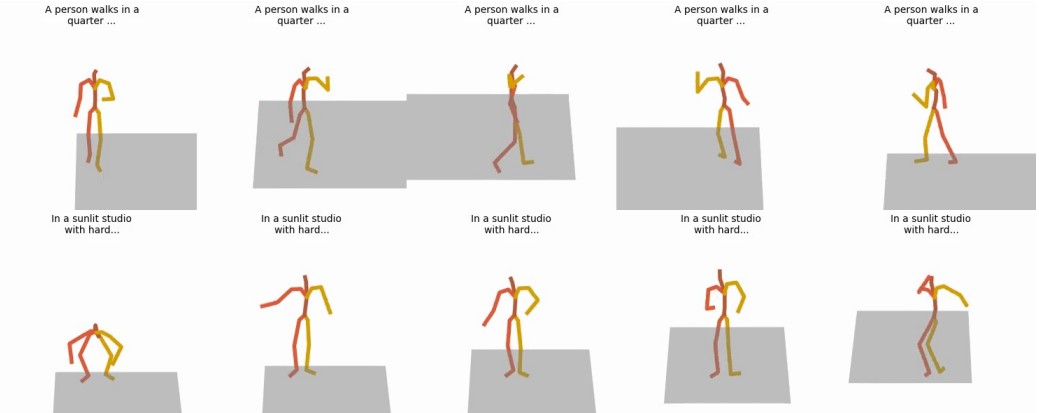

Top: A person walks in a quarter circle to the left, then raises their arm as they continue moving.
Bottom: In a sunlit studio with hardwood floors, ⋯ a young dancer gracefully walks in a quarter
circle to the left, slowly raising their left arm ⋯

Figure 11: **T2M Qualitative Results.** Representative examples showing how T2M models (Dai et al., 2025) behave under different text prompt structures, illustrating their sensitivity to prompt complexity and motion-centric phrasing.

**Motivation for Prompt Disentanglement** Disentangling the text prompt into motion focused and appearance focused components offers benefits beyond preventing the text motion conflict discussed in the main paper. This separation not only contributes to more stable VDM training by providing clearer motion supervision but also improves the reliability of T2M inference. As illustrated in Fig. 11, T2M models trained on datasets such as HumanML3D (Guo et al., 2022) often receive short and motion centered descriptions, which makes them sensitive to complex prompts that mix appearance, scene context, and stylistic cues. By supplying a clean motion oriented prompt to the T2M module and handling appearance descriptions separately, we maintain consistent action interpretation while allowing flexible control over visual attributes in the VDM stage.

## B IMPLEMENTATION / EXPERIMENTAL DETAILS

### B.1 DETAILS OF CAMERA VIEW EXTRACTION AND ALIGNMENT

In the main paper, we describe our camera-view alignment process as an equivariant procedure that computes camera poses $(R_i, T_i)$ from a reference video and projects the generated vertices into the corresponding views. Here, we provide implementation details of how this is realized in practice,

drawing heavily from SMPLest-X (Yin et al., 2025). Concretely, we first apply SMPLest-X to the reference video frames to extract per-frame SMPL meshes, which we denote as $V_{\text{ref}}$. Since SMPLest-X outputs vertices already transformed into the camera coordinate system—where the camera center is defined as the world origin (i.e., $R = I, T = \mathbf{0}$)-we can directly treat the returned meshes as being expressed under this convention. We then adjust the camera intrinsics based on the detected human bounding box: the focal lengths are rescaled to match the box size (width and height), and the principal point is shifted to align with the box location in the original frame. This re-parameterization allows the $V_{\text{ref}}$ to be projected back and overlaid accurately onto the reference frames. We then bring the vertices generated by our T2M model, denoted as $V_{\text{gen}}$, into the coordinate system of $V_{\text{ref}}$ by translating them so that the pelvis joints are placed at the same coordinate. As a result, the renderings of $V_{\text{gen}}$ with the camera poses estimated by SMPLest-X naturally inherit nearly identical viewpoints and camera framing to those of the reference video. This practical implementation ensures that the generated and extracted meshes remain consistent with each other, while preserving the equivariant property of the camera alignment module described in the main text.

## B.2 PROMPT TEMPLATES FOR VLM/LLM

In our experiments, we employed vision-language models (VLMs) InternVL2.5-38B (Chen et al., 2024) and large language models (LLMs) Qwen3-32B (Qwen, 2025) in several key stages.

**Semantic caption extension** First, we used VLMs to recaption the Motion-X dataset. The goal was to enrich missing semantic information such as lab environments or filming conditions that could otherwise degrade video generation quality.

**Motion–Semantic caption split** Second, we applied prompting strategies to split mixed captions into separate components. This allowed us to clearly separate motion-related descriptions from semantic context, so that each could be more effectively used by the corresponding module.

**Benchmark Construction** Finally, we employed LLMs to construct benchmark scenarios. We first generated combinations of actions, and then grounded them in plausible scenarios to produce both motion and semantic captions.

---

**User prompt:**
You are an AI assistant specializing in rewriting video descriptions.
Your task is to split a single, detailed caption into two new, complete captions: a
"motion_caption" and a "semantic_caption".
1. **motion_caption**: A narrative that ONLY describes actions, movements, and dynamic processes. It should read like a story of what is happening.
2. **semantic_caption**: A descriptive narrative that ONLY describes the people, objects, their static attributes, and the setting. It should read like a description of a photograph.
You must return the result ONLY in a JSON object format with the keys "motion_caption" and "semantic_caption".
**Example:**
*Input Caption:* "A woman in a red coat is walking her poodle through a snowy park. The dog is jumping playfully in the snow."
*Output JSON:*
{
    "motion_caption": "A woman is walking her dog through a park as the dog jumps playfully in the snow.",
    "semantic_caption": "The scene features a woman wearing a red coat and her poodle in a snowy park."
}

---

Table 6: Prompts used for caption split.

**User prompt:**
<image>
Based on the image and the given caption, describe additional details that are not covered in the caption.
Keep the original caption in mind, and focus on complementing it, especially with observations about the surrounding environment,
such as whether the scene appears to be in a lab, a controlled space, or a natural setting.
Mention visual cues like lighting, background objects, equipment, or any visible subtitles or on-screen text.
Return your response as a plain paragraph without any lists, bullet points, or markdown formatting.

Table 7: Prompts used for caption extension.

**User prompt:**
You are given a list of actions with their corresponding body parts:
{filtered_action_info}
Generate exactly {batch_size} short action sequences.
Each sequence must:
- Contain 2 or 3 actions only (never more than 3).
- Include at least one action that ends exactly at 7.0 seconds.
- Include some overlap between actions (e.g., one action starts before the previous ends).
- Use only actions from the list above.
- Use the exact body parts associated with each action (do not invent new ones).
Format each action as:
[action description] ## [start time] ## [end time] ## [body part 1] ## [body part 2] ## ...
Separate each sequence by a blank line.
Return only the sequences. No extra explanation, no markdown.
Here is one example:
pick something with the left hand # 1.0 # 3.5 # left arm # spine
wave with both hands # 3.0 # 7.0 # left arm # right arm
Now generate the sequences:

Table 8: Prompts used for action timeline creation.

**User prompt:**
You are given a sequence of human actions with timestamps and involved body parts.
Each line follows this format:
`[action description] # [start time] # [end time] # [body part 1] # [body part 2] # ...`
Based on the actions provided, generate the following three types of captions:
- **motion_caption**: Describe the sequence of body movements clearly and naturally, focusing only on what the person is physically doing. Do not mention timestamps, durations, or specific body parts like "left arm" or "right leg".
- **semantic_caption**: Describe a realistic situation in which this motion could happen. Include relevant details such as the person's clothing, environment, objects involved, and social or emotional context (e.g., "wearing a suit in an office", "in a park during sunset").
- **full_caption**: Write a single, natural sentence or paragraph that blends both motion_caption and semantic_caption into a concise, human-written description of a video scene — something that could guide or inspire video generation. The result should feel confident and fluent, without uncertain phrases like "might" or "possibly". You do not need to include every detail; focus on what feels natural and visually grounded.
Return your output in the following format:
`motion_caption: <your caption>`
`semantic_caption: <your caption>`
`full_caption: <your caption>`
Here is the input sequence:

Table 9: Prompts used for scenario creation.

### B.3 BENCHMARK EXAMPLES

We provide several representative examples from the benchmark used in our experiments. Each entry includes the input caption, the corresponding motion description, and the enriched semantic caption, illustrating the diversity and level of detail present in the benchmark data. The full benchmark will be released publicly.

---

**Benchmark example 01**

"caption": In the early morning light of a sun-dappled forest clearing, a young male park ranger dressed in a khaki uniform and sturdy boots carefully bends on one leg to pick up a discarded water bottle, then swiftly and purposefully throws it into a nearby recycling bin, maintaining balance and focus to keep the trail clean and safe for visitors.

"motion_caption": A person bends down to pick up an object using their right hand while balancing on one leg and engaging their spine, then swiftly throws the object forward with their right hand.

"semantic_caption": A young male park ranger dressed in a khaki uniform and sturdy boots, working on an early morning patrol in a sun-dappled forest clearing, is intent on keeping the trail clean and safe for visitors.

---

**Benchmark example 02**

"caption": A young professional in a sleek navy suit stands in a sunlit, modern glass-walled office at mid-afternoon, holding a phone to his ear as he speaks intently. After listening carefully, he raises his hand and points decisively toward a digital presentation on a nearby screen, emphasizing a key detail during the focused phone call.

"motion_caption": A person holds a phone to their ear with one hand while speaking, then extends that hand forward to point at something.

"semantic_caption": A young professional in a sleek navy suit stands in a modern glass-walled office at mid-afternoon, bathed in natural light. Clad in polished black shoes and a crisp white shirt, he is engaged in a focused phone call near a digital presentation displayed on a nearby screen, emphasizing a key detail during the conversation.

---

**Benchmark example 03**

"caption": A young female violinist in an elegant black gown stands center stage under warm spotlights in a grand concert hall, expertly moving her arms in coordinated strokes as she passionately performs a solo that fills the velvet-lined auditorium with rich, resonant music.

"motion_caption": A person holds an instrument while moving their arms in coordinated, repetitive motions to produce music.

"semantic_caption": A young female violinist dressed in an elegant black gown stands center stage under the warm glow of spotlights in a grand concert hall. The attentive audience seated in the velvet-lined auditorium listens as she immerses herself completely in the music, while rich, resonant notes fill the air.

---

Table 10: Examples of HuMoBench

### B.4 USER STUDY DETAILS

Participants were given the following instructions:

"Overview": This survey is part of a research user study. In each question, participants are shown two videos and asked to decide which video performs better according to two criteria.

"Motion / Action Quality":
- Whether the person's movement accurately and naturally aligns with the textual description.
- Whether the person's body is represented correctly without distortion or unnatural shapes.

"Video / Visual Quality":
- Whether the background, person, clothing, and other details described in the text are accurately and clearly represented.
- Whether the overall scene looks consistent and natural.

"Response Options":
1. The left video looks better.
2. The two videos look similar.
3. The right video looks better.

Thank you for your participation.

Table 11: User study instructions used in our evaluation.

## C   LLM USAGE

The use of large language models in this study was strictly limited to improving grammar and readability. All aspects of the research including ideation, methodological design, data analysis, and interpretation were conducted solely by the authors.

