# OpenReview forum: "Generating Human Motion Videos using a Cascaded Text-to-Video Framework"
_ICLR.cc/2026/Conference — Submitted to ICLR 2026_

### Official Review · Reviewer_RK72 · 2025-10-31

**Soundness:** 3
**Presentation:** 3
**Contribution:** 2
**Rating:** 4
**Confidence:** 4

**Summary:**

This paper studies the same problem as HMTV, which connects a text-to-motion module with a motion-to-video generator. The authors first disentangle the text prompt into a motion prompt and a semantic prompt. The motion prompt is then converted into a motion sequence using a text-to-motion model, which is subsequently rendered as guidance videos. A tailored conditioning strategy is developed for the motion-conditioned video diffusion model (VDM). Camera pose selection is handled by an early denoising stage of a text-to-video model. The results demonstrate both quantitative and qualitative improvements over the vanilla text-to-video baseline and the prior method, HMTV.

**Strengths:**

1. The observation that large body movements are generated earlier, while finer details appear later, is interesting. Based on this observation, the authors design a conditioning strategy that improves performance.

2. The approach of using a text-to-video model to generate a reference video, and then estimating the approximate camera pose from the early frames of the generated human shapes, is also interesting.

3. The paper demonstrates that the generated motion videos can be edited using the SDEdit approach, which represents a meaningful and practical application of the proposed method.

**Weaknesses:**

1. The paper proposes to recaption videos and disentangle the text prompt into two complementary parts: motion caption and semantic caption. However, there is no ablation study that evaluates each part separately. What would the results be if the videos were recaptioned into a single caption instead? In Table 2, the paper compares results with and without text refinement, but the improvement is unclear, and in fact, the motion metrics drop after refinement.
2. There is no ablation study or sufficient explanation regarding the hyperparameter choices and the intuition behind the diffusion timestep sampling. In particular, the rationale for using a truncated normal distribution, the mean reduction, and the increase of standard deviation should be further clarified.
3. The argument in Lines 200–210 and Figure 3a needs revision. As I understand it, both the vanilla training and the proposed method use the same motion but different text prompts as inputs. If the visual conditions are the same, why does the vanilla training fail to capture the fine-grained motion? Does this imply that the generated motion is not aligned with the motion condition?
4. The camera pose selection is based on an existing text-to-video model, rather than being predicted within the proposed pipeline. This raises doubts about the accuracy of the extracted poses. It would be more elegant and meaningful if the camera poses were predicted directly by the model.
5. The quantitative results in Table 1 are not particularly strong, especially on the MovieGen benchmark.
6. The paper should discuss related work that studies a related problem:
Move-in-2D: 2D-Conditioned Human Motion Generation, CVPR 2025.
7. There are no supplementary videos provided, making it difficult to assess the visual quality of the model.

**Questions:**

1. In the motion editing experiments in Figure 6, all results appear to use the same text prompt. Are there any results showing edits using different text prompts?
2. Why is the first-stage text-to-motion model not retrained on the proposed dataset? Would retraining it improve performance?
3. In Figure 4, should HumanVid actually be CamAnimate? Do both HumanVid and Ours use the same motion sequences as input, while VDM is trained on different data and uses a different backbone?

---

> ### Author Response · Authors · 2025-11-19
>
> **W1-1.** There is no ablation study that evaluates each text part separately. What would the results be if the videos were recaptioned into a single caption instead?
>
> **A.** Regarding the question about using a single recaption, we would like to point out the potential issues this would introduce. First, the performance of the text-to-motion (T2M) model would likely degrade.  Most T2M models are trained on relatively simple textual descriptions compared to the richer and more mixed prompts commonly used for video diffusion models. Below are representative examples from HumanML3D, a widely used T2M dataset, that illustrate the simplicity and motion-focused nature of the captions used during training:
> - the man walks in a counterclockwise circle.
> - a person walks backward slowly.
> - a person does a jump.
>
> Since models are trained on such simple and motion-centric descriptions, they struggle when given complex prompts containing appearance and  scene details. They tend to latch onto the dominant motion-related words rather than faithfully capturing the intended action. This often leads to inaccurate motion generation. Examples illustrating this behavior can be found in Appendix Figure 11.
>
> Second, using a single full recaption can introduce semantic conflicts during training. It is important to note that Figure 3a shows outputs from an early checkpoint, not a fully trained model. At this stage, appearance and scene details in a long prompt can interfere with learning the motion-conditioned mapping, making it harder for the model to follow the target motion. Our approach avoids this issue by using semantic-only prompts during training, removing details that could conflict with the motion signal and enabling more stable optimization.
>
> **W1-2.** In Table 2, the paper compares results with and without text refinement, but the improvement is unclear.
>
> **A.** Regarding Table 2, the averaged motion score can be misleading because the two metrics capture different aspects. We reported the average due to space constraints, but the full results in Appendix Table 4 provide a clearer view. As shown there, text refinement yields a clear improvement in motion smoothness, which is the metric most directly tied to temporal stability.The dynamic degree metric reflects only how energetic the motion is, not how correct or stable it is, so it should be interpreted together with the other motion metrics. In addition, subject consistency increases with our refinement strategy. Although listed under appearance metrics, it is closely linked to motion quality and further supports the effectiveness of separating semantic and motion components.

---

> ### Author Response · Authors · 2025-11-19
>
> **W2.** There is no ablation study or sufficient explanation regarding the hyperparameter choices and the intuition behind the diffusion timestep sampling.
>
> **A.**
> We based our hyperparameter and timestep sampling choices on the settings used in AC3D, which also models camera-conditioned motion and therefore provides a reasonable starting point for our architecture. From this baseline, we tailored the configuration to the human-centric setting of our task, adjusting parameters to better handle articulated motion and person-focused dynamics. The rationale behind these design decisions is described in Section L227–L241 of the paper.
>
> In response to the reviewer’s comment, we also conducted a further ablation of the timestep-conditioning strategy, comparing uniform sampling, the hyperparameters used in AC3D, and our proposed truncated-normal setting. Concretely:
> - **Uniform sampling:** timesteps drawn evenly between `t = 0` and `t = 1.0`
> - **AC3D:** truncated normal distribution with mean `t = 0.95` and `std = 0.1`
> - **Ours:** truncated normal distribution with mean `t = 0.9` and `std = 0.2`
>
> For the truncated-normal variants (AC3D and ours), sampling was restricted to the range `t = 0.6` to `t = 1.0`, and inference-time conditioning was also applied only within this interval. In contrast, the uniform strategy is conditioned over the full range from `t = 0` to `t = 1.0`.
>
> | Method     | Aesthetic Quality | Image Quality | Subject Consistency | Background Consistency | Motion Smoothness | Dynamic Degree | Text Alignment |
> |------------|-------------------|---------------|----------------------|-------------------------|--------------------|-----------------|----------------|
> | Uniform    | **0.541**             | *0.610*         | **0.949**               | *0.944*                   | *0.974*             | 0.323     |   0.259  |
> | AC3D       | 0.504             | 0.587     |      0.940           | 0.938                   | **0.976**         | **0.548**           |   **0.262**      |
> | Ours | *0.531*            | **0.619**       | *0.946*          | **0.951**               | 0.971              | *0.457*         |   **0.262** |
>
>  As shown in the table, uniform sampling yields slightly stronger appearance-related scores but suffers from a significant drop in dynamic degree and requires conditioning over the full range (`t = 0` to `t = 1.0`), leading to higher inference-time cost.
>
> In contrast, our truncated-normal setting achieves a more favorable balance between visual quality and motion expressiveness while remaining efficient at inference time. AC3D’s configuration, which is designed specifically to emphasize camera-motion behavior, also avoids full-range conditioning but produces noticeably lower scores overall.
>
> **W3.** The argument in Lines 200–210 and Figure 3a needs revision.
>
> **A.**  It is correct that the vanilla model and our method use the same motion input but different text prompts. Figure 3a, however, does not show results from a fully trained model; it visualizes outputs from an early checkpoint. At this stage, providing a full descriptive prompt introduces semantic details that may conflict with the motion condition, making it harder for the model to converge toward the correct motion pattern. In contrast, our method uses a semantic-only prompt during training, intentionally removing appearance or action details that could interfere with learning the motion-conditioned mapping. Figure 3a illustrates this difference at the same training stage, where our approach already follows the motion more faithfully than the vanilla model.

---

> ### Author Response · Authors · 2025-11-19
>
> **W4.** The camera pose selection is based on an existing text-to-video model, rather than being predicted within the proposed pipeline. This raises doubts about the accuracy of the extracted poses.
>
> **A.** How to design camera pose generation is ultimately a modeling choice with inevitable tradeoffs.
>
> One option would be to directly train a generative or regression model for camera pose given the input text. With sufficiently large and high-quality paired data, this could in principle serve as an upper bound in performance. However, to the best of our knowledge, large-scale datasets containing human-centric dynamic scenes with accurate camera pose annotations do not currently exist. Most available datasets with camera annotations focus on static environments [1, 2] and are difficult to collect, often containing approximation errors from structure-from-motion pipelines such as COLMAP. These issues are amplified in dynamic scenes, where motion, occlusion, and nonrigid deformation make pose estimation even less reliable. As a result, obtaining the high-quality annotations needed to train such a model is extremely difficult, making this approach unlikely to achieve optimal performance in practice.
>
> Given these data limitations, a more feasible direction is to obtain camera poses implicitly rather than training a separate pose prediction model. Our method follows this strategy and offers a practical, resource efficient alternative. The empirical results in our experiments demonstrate that this approach is effective in practice. We therefore believe that our camera pose selection module provides a meaningful solution under current data constraints.
>
> **W5.** The quantitative results in Table 1 are not particularly strong, especially on the MovieGen benchmark.
>
> **A.** We agree that the quantitative differences on MovieGen appear modest. This is partly because several VBench metrics are already saturated, which limits their sensitivity in distinguishing improvements among recent high-quality models. For this reason, we additionally conducted a user study to better capture perceptual differences.
> |                    | **Win** | **Lose** | **Tie** | **p-value** |
> |--------------------|---------|----------|---------|-------------|
> | **Motion Quality** | **0.631** | 0.214    | 0.145   | 0.0026    |
> | **Visual Quality** | **0.545** | 0.257    | 0.186   | 0.0410    |
>
> Participants viewed nine paired videos generated by the baselines (three samples from each baseline) and our approach for the same prompts and evaluated Motion / Action Quality and Video / Visual Quality by selecting win, lose, or tie. Across 44 participants, our method was preferred overall, confirming the effectiveness of the proposed approach. We elaborate on the user study in 4.4 and Appendix B.4.
>
> **W6.**  The paper should discuss related work that studies a related problem: Move-in-2D: 2D-Conditioned Human Motion Generation, CVPR 2025.
>
> **A.** We thank the reviewer for the suggestion. We have added a discussion of Move-in-2D in the related work section.
>
> **W7.** There are no supplementary videos provided.
>
> **A.** We have provided a link to our anonymous project page on the first page of the submission, where all supplementary videos can be viewed. Please check it for the visual results.
>
> *[1] Stereo Magnification: Learning View Synthesis using Multiplane Images*
> *[2] DL3DV-10K: A Large-Scale Scene Dataset for Deep Learning-based 3D Vision*

---

> ### Author Response · Authors · 2025-11-19
>
> **Q1.**  In the motion editing experiments in Figure 6, all results appear to use the same text prompt. Are there any results showing edits using different text prompts?
>
> **A.** We have added additional motion editing examples in Appendix Figure 9. For the same underlying motion sequence, we also generated multiple edited variants using different text prompts, demonstrating that our VDM adapts the motion edits according to the prompt.
>
> **Q2.** Why is the first-stage text-to-motion model not retrained on the proposed dataset? Would retraining it improve performance?
>
> **A.** Retraining the first-stage text-to-motion model on our dataset is certainly possible and may further improve performance. However, our focus is on constructing a coherent pipeline by integrating strong existing modules. In line with our overall modular design philosophy, we aim to show how far off-the-shelf models can be pushed when combined through careful data preparation and unified integration. For this reason, we did not pursue additional T2M retraining.
>
> **Q3.** In Figure 4, should HumanVid actually be CamAnimate? Do both HumanVid and Ours use the same motion sequences as input, while VDM is trained on different data and uses a different backbone?
>
> **A.** Thank you for pointing out the labeling issue. We have updated the label in Figure 4 from HumanVid to CamAnimate for consistency with the manuscript. Both CamAnimate and our method use the same input motion sequences and camera view point for a fair comparison. It is also correct that our VDM is trained on different data and uses a different backbone.

---

> > ### Comment · Reviewer_RK72 · 2025-11-26
> >
> > I appreciate the authors’ detailed response. I would like to increase my score to 6.

---

### Official Review · Reviewer_APoR · 2025-11-01

**Soundness:** 3
**Presentation:** 3
**Contribution:** 2
**Rating:** 4
**Confidence:** 4

**Summary:**

This paper aims at human video generation from text prompts. The proposed method, named CAMEO, is a cascaded T2V generation pipeline, which consists of two parts: T2M and VDM. The T2M component utilizes an off-the-shelf model (STMC) to generate low-dimensional SMPL motion sequences, which are then rendered (conditioned on the camera-view-selection) and fed into a 2D-motion-conditioned video diffusion model to produce the final video. CAMEO is compared with recent methods (HTMV and CamAnimate) and outperforms them in most quantitative evaluation metrics. The paper also presents ablation studies regarding the choice of text refinement strategy and the importance of the view selection module.

**Strengths:**

1. **Clear pipeline**. It connects a text-to-motion module and a motion-conditioned video diffusion model for more robust human motion video generation, which is technically sound.
2. **Good caption design**. The caption re-captioning with motion/semantics split reduces conflicts during VDM training.
3. The Camera view selection idea is simple yet effective, leveraging early denoising results in the text-to-video diffusion model to extract view changes.

**Weaknesses:**

1. The approach is quite similar to HMTV and does not demonstrate a significant conceptual or methodological improvement.
2. The procedure depends on early denoising frames and SMPL estimation; robustness to text domains (e.g., stylized, occlusions) is unclear.
3. The proposed text refinement contributes only marginal improvements, as shown in Table 2; ablation results suggest it may not be a major factor.
4. The base models used in comparison differ, making the claimed superiority less meaningful, as the quality of the base T2V model has a substantial influence on the resulting human motion videos.
5. No user evaluation or human preference test is provided to validate the claimed perceptual improvements.
6. Table 1 performance markings are incorrect -- the “best” and “second-best” indicators do not match the actual numerical values, which weakens result's credibility.

**Questions:**

How robust is the camera module? What happens for stylized or animation-like prompts?

---

> ### Author Response · Authors · 2025-11-19
>
> **W1.** The approach is quite similar to HMTV and does not demonstrate a significant conceptual or methodological improvement
>
> **A.** We acknowledge that some high-level aspects of our pipeline resemble HMTV. However, unlike HMTV, we identify and analyze conflicts that arise when multiple conditioning signals are combined, which were previously overlooked. We clarify why naively stitching modules could lead to unstable behavior and show that resolving such conflicts is essential for constructing a reliable and coherent pipeline.Another key difference is that we introduce a view selection module to automate the camera choice process, whereas HMTV requires users to manually specify camera views. This component is necessary for building a scalable and fully automated pipeline for human centric video generation. Our contribution therefore lies not only in the integration itself but also in exposing these underlying design challenges and demonstrating that addressing them yields a more effective system.
>
> **W2.** The procedure depends on early denoising frames and SMPL estimation; robustness to text domains is unclear.
>
> **A.** Our framework cleanly separates motion from all non-motion factors: while motion is determined by SMPL-based estimation, appearance, environment, and style are fully controlled through the text prompt and can vary independently of the motion. Additional qualitative examples in the Appendix Figure 8 show that the method remains stable under challenging scenarios such as partial occlusions and subjects moving partially out of frame. Moreover, Figure 4 demonstrates that the model handles stylized prompts (e.g., japanese animated), indicating strong robustness to diverse text domains. These observations suggest that the reliance on early denoising frames and SMPL estimation does not hinder generalization across varied textual conditions.
>
>
> **W3.** The proposed text refinement contributes only marginal improvements.
>
> **A.** We would like to clarify that the marginal gains shown in Table 2 are partly due to averaging motion smoothness and dynamic degree. We averaged them because of space constraints, but this can mask the actual effect of the refinement step. As shown in the full metrics provided in the Appendix Table 4, the refinement consistently improves motion smoothness and subject consistency, which are more directly related to temporal stability and coherent articulation.
>
> Dynamic degree, in contrast, measures how energetic the motion is rather than how correct it is, so combining it with smoothness in a single averaged score can reduce the apparent improvement. When considering the full set of metrics, the refinement step plays a meaningful role in producing more stable and consistent motion.
>
> **W4.** The base models used in comparison differ, making the claimed superiority less meaningful, as the quality of the base T2V model has a substantial influence on the resulting human motion videos.
>
> **A.** We would like to clarify that we used the same base model whenever possible to ensure a fair comparison. Specifically, the vanilla CogVideoX-5B serves as the base model for both our tuned version and the tuned HMTV model, since we recognize that the underlying T2V model strongly influences generation quality. CamAnimate is the only exception, as human-centric video baselines are limited and it is one of the few relevant methods available. Other than this necessary exception, we aligned the base models to keep the comparison fair.
>
>
> **W5.** No user evaluation or human preference test.
>
> **A.**
> Following the suggestion, we conducted a user study to validate the effectiveness of our method.
> |                    | **Win** | **Lose** | **Tie** | **p-value** |
> |--------------------|---------|----------|---------|-------------|
> | **Motion Quality** | **0.631** | 0.214    | 0.145   | 0.0026    |
> | **Visual Quality** | **0.545** | 0.257    | 0.186   | 0.0410    |
>
> Participants viewed nine paired videos generated by the baselines (three samples from each baseline) and our approach for the same prompts and evaluated Motion / Action Quality and Video / Visual Quality by selecting win, lose, or tie. Across 44 participants, our method was preferred overall, confirming the effectiveness of the proposed approach. We elaborate on the user study in 4.4 and Appendix B.4.
>
> **W6.** Table 1 performance markings are incorrect -- the “best” and “second-best” indicators.
>
> **A.** We thank the reviewer for pointing out the inconsistency in Table 1. We have corrected the best and second-best indicators to match the numerical values.

---

> ### Author Response · Authors · 2025-11-19
>
> **Q1.** How robust is the camera module? What happens for stylized or animation-like prompts?
>
> **A.** The camera module remains robust even for stylized or animation-like prompts. Since it operates as a video diffusion model, its behavior is largely independent of the visual style. For example, in Figure 4 we use prompts such as ‘A Japanese animated film of a young woman standing on a ship …’ and the module produces stable and consistent camera trajectories. We also note that, across all benchmark evaluations and internal tests, we did not observe any clear failure cases of the camera module, indicating that it is generally reliable across diverse prompt domains.

---

### Official Review · Reviewer_5AhP · 2025-11-02

**Soundness:** 3
**Presentation:** 3
**Contribution:** 2
**Rating:** 6
**Confidence:** 4

**Summary:**

Authors propose a framework to decompose Text to Motion and Video Diffusion Models to generate videos conditioned well on Motion and view points.

Authors bring in techniques (e.g. refinement by LLMs using rendering tools) to make the pipeline continuous end to end with minimal (or no) need for human interruptions from prompts to final video.

A new dataset is also proposed.

**Strengths:**

Comprehensive and Complete pipeline. Authors study the video human centric video generation as an integrated system and do not leave the bottlenecks (like view point conditioning) out of the solution.

The provided qualitative results look promising.

The new dataset brings some more novelty to this work.

Quantitative results are competitive on MovieGen with state of the art, and is mostly the best on the proposed dataset.

**Weaknesses:**

Although I appreciate the completeness of the approach, but there is not much novelty in each element being used and to some extend the method seems like an ad-hoc utilization of some off-the-shelf tools. The VDM controlNet is the only part that is trained by a new condition.

**Questions:**

1- It seems to me that Tab 2, ablation study, is not supporting the contribution of different steps. Any clarification on this?

2- Is there an analysis on visibility of different body parts? I feel in most of the qualitative results, the lower body is not visible. Can this be controlled by m_{1:k}?

3- Is there any measurement on the diversity of the view points in the dataset, and the generated videos?

---

> ### Author Response · Authors · 2025-11-19
>
> **W1.** Although I appreciate the completeness of the approach, but there is not much novelty in each element being used and to some extend the method seems like an ad-hoc utilization of some off-the-shelf tools.
>
> **A.** We acknowledge that our framework builds on several off-the-shelf components and that each module alone does not introduce substantial novelty. The main contribution of our work lies in establishing a coherent end-to-end pipeline through careful data preparation and tight integration of these components. This unified design yields significantly stronger results than using the parts in isolation and highlights how existing tools can be pushed further when combined within a principled framework.
>
> **Q1.** Tab 2, ablation study, is not supporting the contribution of different steps.
>
> **A.**  We tailored four key components in our paper: 1) the captioning process, 2) the visual cue design, 3) the timestep conditioning, and 4) the camera view selection module. Components (1) and (4) are individually ablated in Table 2 and Figure 5. We therefore additionally ablated components (2) and (3).  Due to limited time, all ablation variants were trained only to early epochs. For consistency, the Ours baseline in these ablations also uses the corresponding early-epoch checkpoint. As a result, the reported numbers may differ from those of the fully trained models in the main paper.
>
> For (2), we compared keypoints against our SMPL-based rendering to assess the effect of different visual cues. SMPL-based cues achieved better performance on most metrics.
> | Method     | Aesthetic Quality | Image Quality | Subject Consistency | Background Consistency | Motion Smoothness | Dynamic Degree | Text Alignment |
> |------------|-------------------|---------------|----------------------|-------------------------|--------------------|-----------------|----------------|
> | Keypoint    | 0.527             | 0.610         | **0.947**               | 0.950                   | 0.968              | **0.459**     | 0.259      |
> | Ours | **0.531**            | **0.619**       | 0.946           | **0.951**              | **0.971**              |  0.457          |    **0.262** |
>
> For (3), we evaluated multiple timestep-conditioning strategies, including uniform sampling, the hyperparameters used in AC3D, and our proposed setting. Concretely:
> - **Uniform sampling:** timesteps drawn evenly between `t = 0` and `t = 1.0`
> - **AC3D:** truncated normal distribution with mean `t = 0.95` and `std = 0.1`
> - **Ours:** truncated normal distribution with mean `t = 0.9` and `std = 0.2`
>
> For the truncated-normal variants (AC3D and ours), sampling was restricted to the range `t = 0.6` to `t = 1.0`, and inference-time conditioning was also applied only within this interval. In contrast, the uniform strategy is conditioned over the full range from `t = 0` to `t = 1.0`.
>
> | Method     | Aesthetic Quality | Image Quality | Subject Consistency | Background Consistency | Motion Smoothness | Dynamic Degree | Text Alignment |
> |------------|-------------------|---------------|----------------------|-------------------------|--------------------|-----------------|----------------|
> | Uniform    | **0.541**             | *0.610*         | **0.949**               | *0.944*                   | *0.974*             | 0.323     |   0.259  |
> | AC3D       | 0.504             | 0.587     |      0.940           | 0.938                   | **0.976**         | **0.548**           |   **0.262**      |
> | Ours | *0.531*            | **0.619**       | *0.946*          | **0.951**               | 0.971              | *0.457*         |   **0.262** |
>
> Although uniform sampling achieves comparable results and is slightly better in some appearance and motion metrics, it also shows a clear drop in dynamic degree, indicating reduced motion expressiveness. In addition, because it conditions over the entire timestep range (`t = 0` to `t = 1.0`), it requires applying conditioning across many more steps and therefore incurs higher inference-time cost. AC3D’s truncated-normal strategy avoids uniform sampling’s full-range conditioning but yields noticeably lower scores. In contrast, our truncated-normal setting maintains strong overall video quality while preserving motion expressiveness, and does so with greater inference-time efficiency.

---

> ### Author Response · Authors · 2025-11-19
>
> **Q2.** I feel in most of the qualitative results, the lower body is not visible. Can this be controlled by $m_{1:k}$?
>
> **A.** Yes. The visibility of the lower body can be controlled through $m_{1:k}$. This can be done either automatically using our proposed view selection module or manually by specifying the desired camera view. This explicit controllability is one of the strengths of our pipeline. Since our method focuses on human centric video generation, the qualitative results tend to favor upper body views. We include additional examples with visible lower bodies in the Appendix Figure 8.
>
> **Q3.** Is there any measurement on the diversity of the view points in the dataset, and the generated videos?
>
> **A.** To evaluate the diversity of camera viewpoints, we extracted the 3D camera translation vectors from the reference videos that we generated specifically for determining camera views. For the 120 HumoBench clips, each reference video provides a single translation vector of dimension (3) extracted from its first frame. Appendix Figure 10 presents the corresponding scatter plot and statistics, showing a wide spread across all three axes and demonstrating that the generated reference views span a broad range of camera positions rather than collapsing to a narrow region. Qualitative examples also show diverse viewpoints, including upper-body close-ups, full-body shots, and lower-angle views such as shin-level perspectives.

---

### Official Review · Reviewer_XEUb · 2025-11-04

**Soundness:** 2
**Presentation:** 2
**Contribution:** 2
**Rating:** 4
**Confidence:** 5

**Summary:**

the paper proposes a text-to-motion augmented cascaded text-to-video generation framework for human motion video generation

**Strengths:**

- the paper proposes a feasible solution for reliable human motion generation in video generation models
- the proposed framework clearly improves the correctness of generated human body structures and fidelity of generated poses and motions
- extensive experiments demonstrate the effectiveness of the proposed modules and the effectiveness of semi-explicit 3D controls

**Weaknesses:**

- while the paper demonstrates improvements in terms of single-person motion for the video generation task, the novelty of the proposed framework seems to be limited: the proposed framework feels like a combination of existing components, which are added up together to solve a specific and narrowed-down problem. while the paper acknowledged the limitations of generalizability, it still unclear how robust the proposed framework is when handling more complicated cases for single-person scene video generation, e.g., what happens when camera distance significantly changes, or can the proposed framework handle the cases where the person is occluded or partially out of the frame or temporally missing in some frames
- while the paper adopted VBench for quantitative comparisons, the reliability of the VBench metrics is still not well justified. based on the reviewer's experience, some scores might favor specific aspects of videos while ignoring the actual visual quality. it is highly recommended to conduct a user study to validate the effectiveness of the proposed method considering the human evaluation is still the most reliable metric for video generation tasks
- the quality of demonstrated applications of motion editing and camera view editing seem to be not good in terms of subject consistency

**Questions:**

please refer to weaknesses section

---

> ### Author Response · Authors · 2025-11-19
>
> **W1.** The novelty of the proposed framework seems to be limited: the proposed framework feels like a combination of existing components, which are added up together to solve a specific and narrowed-down problem.
>
> **A.** We acknowledge that our framework employs several off-the-shelf components and that each individual module does not introduce major novelty. However, unlike prior work, we identify and analyze conflicts that arise when multiple conditioning signals are combined, which were previously overlooked. We clarify why naively stitching modules could lead to unstable behavior and show that resolving such conflicts is essential for constructing a reliable and coherent pipeline. Our contribution therefore lies not only in the integration itself but also in exposing these underlying design challenges and demonstrating that addressing them yields a more effective system. Looking ahead, multi condition designs will remain important for user controllability, and the conflicts we highlight are intrinsic to this setup. Understanding these challenges will continue to provide lasting value for future research and system design.
>
> We also stress that the problem we target is intentionally focused, because human motion generation remains a critical bottleneck for general video generation rather than a niche application. As discussed in our manuscript L54-L56, and with the corresponding citations newly added, domains such as robotics, film production, digital avatars, and fashion all require accurate and controllable human motion synthesis. Tackling this focused yet fundamental challenge is therefore essential for advancing broader video generation capabilities.
>
> Moreover, while our goal in this proposal is a specific problem, we believe the implications extend beyond what is directly shown in the paper. In robotics, for example, modern video world models such as Cosmos-Transfer2.5 [1] increasingly rely on compact visual inputs, including manipulator renderings or segmentation maps, reflecting the growing need for controllability. Yet there is limited research on how such low-dimensional conditioning signals should be generated or whether a modular generative approach is effective. Our work provides a concrete example of this direction. In this sense, CAMEO represents a special case of a broader class of modular generative models and serves as an initial proof-of-concept.
>
> **W2.** limitations of generalizability; what happens when camera distance significantly changes, or can the proposed framework handle the cases where the person is occluded or partially out of the frame or temporally missing in some frames
>
> **A.** We have expanded our qualitative analysis and included additional challenging cases in Appendix Figure 8. These examples show that our framework remains stable under various camera view changes, including shifts in distance, as well as scenarios with partial occlusions and subjects moving partially out of frame, demonstrating strong generalization across diverse conditions.
>
> **W3.** it is highly recommended to conduct a user study.
>
> **A.** Following the suggestion, we conducted a user study to validate the effectiveness of our method.
> |                    | **Win** | **Lose** | **Tie** | **p-value** |
> |--------------------|---------|----------|---------|-------------|
> | **Motion Quality** | **0.631** | 0.214    | 0.145   | 0.0026    |
> | **Visual Quality** | **0.545** | 0.257    | 0.186   | 0.0410    |
>
> Participants viewed, in a blind manner, nine paired videos generated by the baselines (three samples from each baseline) and our approach for the same prompts. These video pairs were selected from the examples shown in the main paper and the project page. Participants then evaluated Motion / Action Quality and Video / Visual Quality by choosing win, lose, or tie. Across 44 participants, our method was preferred overall, confirming the effectiveness of the proposed approach. We elaborate on the user study in 4.4 and Appendix B.4.
>
> **W4.** the quality of demonstrated applications of motion editing and camera view editing seem to be not good in terms of subject consistency.
>
> **A.** We agree that the subject is not perfectly identical in the demonstrated applications. This outcome is expected, as these examples were intended only to demonstrate that such applications can be supported within the same unified pipeline with minimal additional mechanisms. Achieving fully consistent identity is a separate objective because diffusion models inherently introduce stochastic variation in appearance even under identical text prompts, and maintaining identity stability typically requires additional identity-preserving modules. Addressing this aspect lies outside the scope of our feasibility oriented demonstration.
>
> *[1] Ali, Arslan, et al. "World Simulation with Video Foundation Models for Physical AI."*

---

### Author Response · Authors · 2025-11-19
**General Response**

We thank the reviewers for their constructive, positive, and thorough reviews. We are happy that the reviewers think that our paper **presents a clear and effective pipeline** (`XEUb`, `5AhP`, `APoR`, `RK72`), offers simple yet impactful design choices, and **provides competitive and promising results supported by extensive experiments** (`XEUb`, `5AhP`). Below, we first summarize the main concerns raised in the reviews and then provide detailed point by point responses.

**1. Limited Novelty** (`XEUb`, `5AhP`, `APoR`)

**A.** While we acknowledge that our framework uses several off the shelf components, we would like to emphasize that our main contribution lies in **identifying and addressing fundamental design factors that are critical for reliable human centric video generation**.
We identify overlooked dual conditioning conflicts that arise when multiple signals jointly guide the generation process, yet have not been examined in prior work despite their central role in achieving stability, controllability, and consistency in human centric generation. By explicitly analyzing these conflicts and introducing our view selection module, we show that resolving such interactions is essential for stable and controllable behavior.

Although our analysis focuses on visual cue and text conditioning, the underlying principles apply more broadly to systems where structured conditioning signals must be generated or orchestrated. For example, recent world models such as Cosmos Transfer2.5 [1] rely on multiple conditoning signals to control behavior, yet the process of generating and harmonizing these signals remains underexplored. In this sense, our framework provides a concrete case study and an initial proof of concept for **modular generative models where control arises from the careful design and coordination of structured signals**. We expect that our findings can inform future research in this line of work by motivating more principled approaches to creating and integrating conditioning signals.

**2. Conducting User study** (`XEUb`, `APoR`, `RK72`)

**A.** Several reviewers pointed out that our method does not appear clearly superior under standard metrics. While VBench is widely used, many of its scores are known to saturate and can be insensitive to nuanced differences in human centric video quality. To address this limitation and provide a more reliable assessment, we conducted an additional user study upon request.

|                    | **Win** | **Lose** | **Tie** | **p-value** |
|--------------------|---------|----------|---------|-------------|
| **Motion Quality** | **0.631** | 0.214    | 0.145   | 0.0026    |
| **Visual Quality** | **0.545** | 0.257    | 0.186   | 0.0410    |

Participants blindly compared paired videos from the baselines and our method, judging motion quality and visual quality using win, lose, or tie. A total of 44 participants took part in the study. **Our method achieved more wins than the baselines on both criteria, with statistically significant differences** based on a binomial test excluding ties (motion: p=0.0026, visual: p=0.0410). We elaborate on the user study in 4.4 and Appendix B.4.

**3. Generalization Ability** (`5AhP`, `APoR`)

**A.** Several reviewers asked about the generalization ability of our method in more challenging scenarios, such as drastic camera view changes or cases where the person becomes partially out of frame. To address these questions, we added additional qualitative results covering a wider range of extreme conditions, which are now included in Appendix Figure 8. These examples illustrate that **our pipeline remains stable and coherent even under such challenging variations**.

*[1] Ali, Arslan, et al. "World Simulation with Video Foundation Models for Physical AI."*

---

### Meta-Review · Area_Chair_fP6g · 2026-01-07

**Summary:**

The AC carefully reviewed the paper and the full discussion. The submission received mixed initial scores (4, 4, 6, 4). Reviewers generally agreed that the paper presents a feasible approach to improving reliability in human motion generation for video models. They found that the framework enhances anatomical correctness and better preserves pose and motion fidelity, and the extensive experiments support the effectiveness of the proposed modules, including the use of semi-explicit 3D controls.

However, the primary concerns relate to limited novelty: the method is largely perceived as an integration of existing components assembled to address a relatively narrow problem setting. Although the paper acknowledges limited generalizability, it remains unclear how robust the approach is in more complex single-person video scenarios. Reviewers also noted weak visual quality, including issues with subject consistency, which further raises concerns about robustness, generality, and practical usability. Overall, it is uncertain whether the proposed method and pipeline provide clear, meaningful advantages for current or future human video generation models. Given that the discussion is unlikely to resolve these core issues and the scores trend toward rejection, I am inclined to recommend rejection.

**Reviewer Concerns:**

Some concerns—such as the lack of human evaluation or a user study (raised by XEUb, APoR, RK72) and questions about generalization ability (raised by 5AhP and APoR)—have been partially addressed. However, several non-negligible issues remain.

Three reviewers (e.g., XEUb, 5AhP, APoR) pointed to limited novelty. In particular, the proposed framework is largely perceived as an aggregation of existing components assembled to address a specific, relatively narrow problem. Although the paper acknowledges generalizability limitations, it is still unclear how robust the framework is when applied to more complex single-person video generation scenarios.

There are also concerns about insufficient visual evidence. While the authors provide a link to results, it contains fewer than 10 videos, which is generally inadequate for thorough qualitative assessment. Moreover, although the method improves human motion quality, reviewers noted that it may degrade overall aesthetic quality and subject consistency.

Overall, it remains unclear whether the proposed approach provides clear or meaningful benefits for current or future human motion generation models.

**Reviewer Scores:**

Reviewer XEUb is likely to maintain a negative score, primarily due to concerns about limited novelty and weak visual quality.

Reviewer 5AhP is expected to keep the current score unchanged.

Reviewer APoR will likely retain a negative score, mainly citing the novelty issue.

Reviewer RK7R is also likely to increase their score.

---

### Decision · Program_Chairs · 2026-01-26

Reject